# Coxsackievirus A6 Recombinant Subclades D3/A and D3/H Were Predominant in Hand-Foot-And-Mouth Disease Outbreaks in the Paediatric Population, France, 2010–2018

**DOI:** 10.3390/v14051078

**Published:** 2022-05-17

**Authors:** Stéphanie Tomba Ngangas, Maxime Bisseux, Gwendoline Jugie, Céline Lambert, Robert Cohen, Andreas Werner, Christine Archimbaud, Cécile Henquell, Audrey Mirand, Jean-Luc Bailly

**Affiliations:** 1Université Clermont Auvergne, LMGE CNRS 6023, UFR de Médecine et des Professions Paramédicales, 63001 Clermont-Ferrand, France; stephanie.dan@aphp.fr (S.T.N.); mbisseux@chu-clermontferrand.fr (M.B.); gwendoline.jugie@uca.fr (G.J.); carchimbaud@chu-clermontferrand.fr (C.A.); chenquell@chu-clermontferrand.fr (C.H.); amirand@chu-clermontferrand.fr (A.M.); 2CHU Clermont-Ferrand, Centre National de Référence Des Entérovirus et Parechovirus, Laboratoire de Virologie, 63003 Clermont-Ferrand, France; 3CHU Clermont-Ferrand, Service Biométrie et Médico-Economie—Direction de la Recherche Clinique et Innovation, 63003 Clermont-Ferrand, France; clambert@chu-clermontferrand.fr; 4Association Clinique et Thérapeutique Infantile du Val de Marne (ACTIV), 94000 Créteil, France; robert.cohen@activ-france.fr; 5Association Française de Pédiatrie Ambulatoire (AFPA), 45000 Orléans, France; aw30400@me.com

**Keywords:** atypical hand-foot-and-mouth disease, enterovirus surveillance, clinical epidemiology, molecular typing, third-generation sequencing, whole-genome sequencing, recombination

## Abstract

Coxsackievirus A6 (CVA6) emerged as the most common enterovirus of seasonal outbreaks of hand-foot-and-mouth disease (HFMD). We investigated CVA6 genetic diversity among the clinical phenotypes reported in the paediatric population during sentinel surveillance in France between 2010 and 2018. CVA6 infection was confirmed in 981 children (mean age 1.52 years [IQR 1.17–2.72]) of whom 564 (58%) were males. Atypical HFMD was reported in 705 (72%) children, followed by typical HFMD in 214 (22%) and herpangina in 57 (6%) children. Throat specimens of 245 children were processed with a target-enrichment new-generation sequencing approach, which generated 213 complete CVA6 genomes. The genomes grouped within the D1 and D3 clades (phylogeny inferred with the P1 genomic region). In total, 201 genomes were classified among the recombinant forms (RFs) A, B, F, G, H, and N, and 12 genomes were assigned to 5 previously unreported RFs (R–V). The most frequent RFs were A (58%), H (19%), G (6.1%), and F (5.2%). The yearly number of RFs ranged between 1 (in 2012 and 2013) and 6 (2018). The worldwide CVA6 epidemic transmission began between 2005 and 2007, which coincided with the global spread of the recombinant subclade D3/RF-A.

## 1. Introduction

Human diseases associated with coxsackievirus A6 (CVA6) were rarely reported in European countries before the occurrence of two outbreaks of hand-foot-and-mouth disease (HFMD) in the paediatric populations of Finland (2008) and Spain (2010–2012) [1,2]. After these two outbreaks, several CVA6 upsurges were reported worldwide [3,4,5,6,7]. CVA6 was mainly involved in sporadic disease cases of herpangina (HA), but mounting evidence points to its capacity to cause large seasonal outbreaks of HFMD [8] associated with atypical presentations including vesiculobullous eruption [9,10,11], adult cases [12,13,14], and neurological manifestations [15]. The reasons why the virus was not reported in significant outbreaks until the beginning of the 21st century are unknown [16]. The recent global emergence of CVA6 as the main cause of worldwide HFMD outbreaks and the change in clinical phenotype from HA to atypical HFMD, suggests a shift in transmissibility, antigenicity, or overall genetic features. CVA6 was classified into four clades designated A–D and three subclades D1–D3, based on the phylogenetic analysis of the VP1 protein gene sequences [17]. Earlier molecular epidemiology studies suggested that recent HFMD outbreaks were associated with distinct CVA6 lineages, which emerged within the main clade D3 [18,19,20]. Analyses of viral genomes revealed the existence of a large array of recombinant subclades (recombinant forms, RFs) among CVA6 circulating populations [21,22,23].

The aim of the present study is to describe the genetic diversity of CVA6 associated with the clinical phenotypes of 981 CVA6 infection cases reported by primary care paediatricians involved in the prospective sentinel surveillance of HFMD and HA in France between 2010 and 2018. To gain a better understanding of the CVA6 genetic evolution, we developed whole-genome sequencing (WGS) through targeted amplification of the complete viral genome followed by next-generation sequencing. We describe 213 CVA6 genomes, the largest dataset of full-length genomes determined from clinical specimens. Our comparative data revealed two predominant recombinant subclades (D3/RF-F and D3/RF-H) and nine minority subclades in the children population.

## 2. Materials and Methods

### 2.1. Collection of Clinical Specimens during Sentinel Surveillance

All the CVA6 positive specimens analysed in the study were obtained during ambulatory surveillance of children by primary care paediatricians involved in sentinel networks between 2010 and 2018. The first network was deployed for the surveillance of the paediatric population of the Clermont-Ferrand (France) urban area between 1 April 2010, and 31 December 2013 [24]. The second network was set up at the national level since 1 April 2014 [25]. This nationwide network was placed with the support of the PARI (Pédiatrie Ambulatoire et Recherche en Infectiologie; https://afpa.org/; (accessed on 26 March 2022) observatory since May 2017. Sentinel paediatricians collected throat or buccal swabs from children clinically diagnosed with HFMD or HA. Samples, along with a standardised and anonymised clinical form, were sent to the National Reference Centre for enteroviruses and parechoviruses (Clermont-Ferrand, France) for EV testing and identification. Detection of EV RNA and EV molecular typing (based on VP1 protein gene Sanger sequencing) were performed as previously described [25,26]. The clinical manifestations were classified in three groups, based on the clinical items collected. HA alone was defined as predominant ulcerations on the posterior part of the oral cavity and no cutaneous signs. Typical HFMD was defined by the presence of at least two of the typical HFMD localisations, i.e., oral ulcerations, eruptions on the palms, soles, buttocks, knees or elbows excluding any other localisations; and atypical HFMD, by HFMD with the presence of exanthema on non-typical sites or a generalised eruption. Anonymised biological material and data were used for all analyses.

### 2.2. Primer Design and cDNA Synthesis for Whole-Genome Amplification

An alignment was constructed with 922 complete genome sequences representing all EV-A types and genogroups publicly available in the National Center for Biotechnological Information (NCBI) GenBank (accessed on March 2019). The alignment was used to identify short stretches sharing a minimum of 90% nucleotide identity within in the 5′ and 3′ noncoding regions and to construct oligonucleotide primers for cDNA synthesis and genome amplification. Clinical CVA6 positive specimens were subjected to extraction of total nucleic acids with the NucliSENS easyMAG semi-automated system (bioMérieux, Marcy-l’Étoile, France). The nucleic acids were recovered in 50 μL of elution buffer. Immediately after nucleic acids extraction, reverse transcription was performed with 5 μL of template RNA and the superscript IV reverse transcriptase as recommended by the manufacturer (Invitrogen, Cergy-Pontoise, France). cDNA products were processed immediately for complete genome amplification or stored at −20 °C for subsequent use. The genome amplification reaction was created with 2 µL of cDNA and 18 µL of a reaction mix containing 10 µL of 2X Platinum SuperFi (Invitrogen, Cergy-Pontoise, France), 1 µL of each primer solution (10 µM), and 6 µL of nuclease-free water. The following PCR conditions were used: 1 cycle of 1 min at 98 °C, 41 cycles of 5 s at 98 °C, 10 s at 66 °C, and 4 min 30 s at 72 °C, and a final cycle of 5 min at 72 °C. To confirm a size higher than 7 kb, PCR products were controlled using 1% agarose gel electrophoresis stained with GelStarTM Nucleic Acid Gel Stain (Lonza, Basel, Switzerland).

### 2.3. Construction of Genomic Libraries and Library Sequencing

Single-molecule real-time long-reads sequencing was performed with a Pacific Biosciences Sequel Sequencer (Pacific Biosciences, Menlo Park, CA, USA). The SMRTBell library was prepared using a DNA Template Prep Kit 1.0, following the procedure for preparing Amplicon Libraries using PacBio Barcoded Adapters for multiplex SMRT Sequencing. The PCR products were purified using 0.45X AMPure PacBio beads and normalized to a 20 ng/µL concentration. A Fragment Analyzer (Agilent Technologies, Santa Clara, CA, USA) assay was used to assess the fragment size distribution. Each PCR product were subjected to one-step enzymatic end-repair and blunt-end ligation of barcoded adapters. After equimolar pooling of the 96 barcoded amplicons, 0.45X AMPure PB beads was used for size-selection of fragments above 3 kb. In total, 5 µg of the pooled amplicons was subjected to DNA damage repair followed by exonuclease treatment to generate the SMRTBell template. The SMRTBell library quality was assessed with a Fragment Analyzer System (Agilent Technologies, Santa Clara, CA, USA) and was quantified with a Qubit fluorimeter and Qubit dsDNA HS reagent Assay kit (Life Technologies, Carlsbad, CA, USA). A ready-to-sequence SMRTBell polymerase complex was generated using Binding Kit 3.0 and the primer V4 (Pacific Biosciences, Menlo Park, CA, USA), the diffusion loading protocol was used, according to the manufacturer’s instructions. The library was loaded by the diffusion method at an on-plate concentration of 8 pM of DNA with the Sequel I System. Three libraries with barcodes were loaded onto the PacBio Sequel sequencer in three independent runs into a single SMRT cell. The PacBio Sequel instrument was programmed to load and sequence the sample on PacBio SMRT cells v2.0 (Pacific Biosciences, Menlo Park, CA, USA), acquiring one movie of 20 h per SMRTcell.

### 2.4. Bioinformatic Pipeline for CVA6 Genome Analysis

In the assembly workflow, the raw PacBio reads were assembled using the SMRT Analysis software (Pacific Biosciences, Menlo Park, CA, USA) and the management of sequence analyses was configured through a predefined settings file. The full-length CVA6 genomes were generated with the tools included in SMRT Link^®^ software v6.0.1 (SMRT-Tools-Reference-Guide-v8.0, Pacific Bioscience, Menlo Park, CA, USA). Final consensus sequences for CVA6 identified in each sample was generated based on an 80% similarity threshold and then checked manually for alignment errors. The 213 CVA6 genomes determined in the present study were deposited in GenBank under the accession numbers MT814404–MT814616.

### 2.5. Phylogenetic Analyses of Complete Genomes and Open Reading Frames (ORF)

Maximum likelihood (ML) phylogenies were inferred with a general time reversible (GTR) nucleotide substitution model, a gamma distribution of rate variation among nucleotide sites, and a proportion of invariant sites (GTR + Γ + I) implemented in the PhyML software [27,28]. Statistical support of tree nodes was assessed by a bootstrap procedure with 1000 replicates. Four sequence datasets covering different genomic regions were explored to determine which genomic region was suitable for constructing a time-resolved phylogeny: the P1 region (encoding capsid proteins), the P2 and P3 (encoding non-structural proteins), and the 3D polymerase (3Dpol) gene. Molecular clock was analysed with the program TempEst that plots the genetic distances calculated between the root and tips of a ML phylogeny against the sampling dates [29]. Temporal phylogenies were reconstructed for the P1 region and 3Dpol gene with the Bayesian Markov chain Monte Carlo (BMCMC) method implemented in BEAST 2 [30]. The phylogenies were inferred with the GTR + Γ + I model, a relaxed molecular clock model with uncorrelated lognormal distribution across lineages [31], and a Bayesian skyline coalescent model [32]. Calculations were run with chain lengths of 400 million to ensure convergence of MCMC chains; four runs were combined with the logCombiner software v1.10 [30]. Marginal posterior distributions of parameters were calculated (burnin = 20%) with Tracer v1.7 [33]. The maximum clade credibility (MCC) trees were generated with the program TreeAnnotator (burnin = 20%) from the posterior distribution of trees. The geographic spread of CVA6 was explored with an automated phylogenetic approach using Nextstrain Augur toolkit (https://docs.nextstrain.org/projects/augur/en/stable/index.html; accessed on 1 April 2020) [34] and was applied to the P1 genome region. The Nextstrain pipeline generates an interactive visualization integrating phylogeny and metadata such as the geographic location of clinical samples used to obtain genomic data.

### 2.6. Recombination Analyses

To compare the genetic diversity among whole CVA6 genomes, we used the DNASP program [35] and 100-nucleotide window sliding-window approach and a step size of 10 nucleotides. For each window, the mean pairwise difference was calculated by dividing the total pairwise nucleotide differences by the total number of comparisons; sequence gaps were ignored.

The recombinant pattern of the 213 CVA6 complete genomes determined in this study was investigated with the following approach. The SimPlot v3.5.1 software [36] was used to compare CVA6 genomes with reference genomes previously assigned to recombinant forms (RFs) A to J [21,22,23]. The pairwise nucleotide similarity between these genomes was calculated with a 200-nucleotide sliding window along the alignment (steps of 20 nucleotides). The pattern of similarity plots allowed us to assign each CVA6 genome to a known or a new RF.

### 2.7. Statistical Analysis

Statistical analysis was performed with Stata software (version 15; StataCorp, College Station, TX, USA). All tests were two-sided, with a Type I error set at 0.05. Categorical variables were expressed as frequencies and associated percentages, and the age as median and interquartile range. Comparisons between independent groups were made with the chi-square test or the Fisher exact test for categorical variables and with the Mann–Whitney test for age.

## 3. Results

### 3.1. Epidemiologic and Clinical Characteristics of HFMD Cases Associated with CVA6 in France

Between 1 January 2010 and 31 December 2018, CVA6 accounted for 4.4% (*n* = 1159/26,063) of all the EV infections reported by the French Network for Enterovirus Surveillance (https://cnr.chu-clermontferrand.fr; accessed on 26 March 2022). The peak of global CVA6 infections reported in 2014 coincided with the nationwide extension of the sentinel network for HFMD. CVA6 was predominant (981 cases, 50.4%) among the 1946 EV-associated HFMD cases reported during the ambulatory surveillance. The detection rate varied from 24% (16/67) in 2013 to 64% (55/86) in 2015 (Figure 1). It is worth noting that nationwide HFMD sentinel surveillance was interrupted between 1 April 2015 and 30 April 2017. Clinical data were available for 99% (976/981) of the children seen in consultation (Table 1). The mean age of patients that tested positive for CVA6 was 1.5 years (range, 1.05–2.21). Manifestations were classified into three categories based on the information on lesion locations reported by paediatrician: 214 (22%) children had typical HFMD, 705 (72%) had atypical HFMD, and 57 (6%) had herpangina. HFMD was associated with herpangina in 483/926 (52%) of cases. The proportion of atypical HFMD manifestations increased from 48% (19/40) in 2010 to 82% (167/204) in 2018, with a peak of 89% (42/47) in 2016, the variations being significant from 2016 onwards compared with the years 2010 and 2012 (*p* < 0.001).

### 3.2. Sequencing of CVA6 Complete Genomes from Clinical Samples

For whole-genome sequencing investigations, we selected clinical specimens based on phylogenetic and epidemiologic features. First, we chose one specimen among those for which the VP1 sequences had ≥99% nucleotide similarity with one another. Second, we selected the specimens based on the relative proportion per year, the assigned subclade, and disease category. Overall, we analysed 245 clinical specimens collected in 180 (74%) patients with atypical HFMD, 47 (19%) with HFMD, and 18 (7%) with herpangina alone. Among the specimens selected, 4% and 96% were assigned to subclades D1 and D4, respectively. Our targeted RT-PCR assay amplified the complete viral genomes directly in 224/245 (91%) specimens. The Ct values of the EV diagnostic RT-PCR test (available for 190 specimens) ranged between 15 and 37.8 (median 25). Whole-genome sequencing using long-read single-molecule sequencing on the PacBio platform was performed in three independent libraries, and overall, we obtained 13,499,739 subreads (Table 2). After filtering for full-length subreads and demultiplexing, the number of circular consensus sequences (CCSs) generated was 108,510 (library 1), 88,608 (library 2), and 171,991 (library 3). Subreads were filtered on a length of 7500 bp and nucleotide sequences containing the 5′ and 3′ noncoding regions. The mean number of complete genome sequences recovered per clinical specimen was 1066 (library 1), 865 (library 2), and 1709 (library 3). The depth of sequence coverage was sufficient for the determination of 213 CVA6 complete genomes.

### 3.3. Genetic Divergence of Whole Genome Sequences and Assignment to Recombinant Forms

Of 213 genomes, 10 (5%) were assigned to the previously reported subclade D1 and 203 (95%) to subclade D3. The genomes shared 86–100% nucleotide similarity (mean = 92%). The genetic divergence within and between the two subclades was lower in the P1 genomic region encoding the structural proteins VP1–VP4 than in the P2 and P3 regions encoding the noncoding proteins 2A–2C and 3A–3D (Figure 2), the highest diversity being observed in the 2B gene (nucleotide positions 3827–4123). The genetic divergence pattern between the non-structural and structural regions suggested the presence of recombinant genomes within our dataset.

The CVA6 genomes were classified into 16 clades designated as RFs A–Q based on the phylogenetic pattern assessed for the 3Dpol gene [21,22,23]. Figure 3 shows a ML phylogeny inferred with 760 3Dpol gene sequences derived from the 213 genomes of our study and 547 genomes available in GenBank. RF-A was assigned to 483/760 (64%) sequences and RFs B to Q were assigned to 262/760 (34%) sequences. The remaining 15 (2%) sequences were distributed across 7 phylogenetic clades, which were classified as new RFs. A subset of 12 sequences determined in the present study were distributed among the following RFs: R (*n* = 2 sequences), S (*n* = 1), T (*n* = 2), U (*n* = 3), and V (*n* = 4). RF-R also included a CVA6 genome (accession number LR027552.1) from our previous study investigating the genetic origins of the emerging C1-like EV-A71v2015 [37]. Finally, two CVA6 genomes available in GenBank (accession numbers MF578324.1 and MF578379.1; Vietnam; [19]) formed the additional RFs, W and X. RF-A and RF-H were predominant in our patient population (Table 3). The proportion of RF-A increased from 47% in 2010 to 100% in 2012 and decreased gradually to 21% in 2018. RF-H was reported for the first time in 2014 among 8% of the specimens and represented 44–50% of those tested between 2016 and 2018. The yearly frequency of the other RFs (B, F, G, N, and R to V) ranged between 4% and 53%.

### 3.4. Distribution of Recombinant Forms among CVA6 Clades and Subclades

Among the five CVA6 clades A–E reported earlier [23], clades B and C were defined on the sole analysis of the VP1 gene sequences, and thus were not present in our phylogenetic reconstructions (Figure 4 and Figure 5). ML phylogenetic trees were inferred from different partitions of the dataset of 760 genomes and were used to calculate a regression of genetic distances between the tips and the best-fitting root against sampling times. The analysis showed that the P1 genomic region was suitable for molecular clock analysis (regression coefficient, R^2^ = 0.64) while the P2 and P3 genomic regions and the 3Dpol gene were unsuitable (R^2^ < 0.2). A dated phylogeny was reconstructed with the 760 P1 nucleotide sequences and the distribution of RFs was analysed across the tree (Figure 4). The majority of CVA6 were classified in clade D. The D1 clade sequences were assigned to RFs B, G, and T, and the sequences within clade D2 were distributed among RFs C, D, and K (Table 3). The D3 sequences clustered into subclades D3.1–D3.3 and were assigned to 16 RFs. The D3.1 sequences (assigned to RF-E) were directly linked to the subclade root, suggesting an ancestral recombinant lineage (D3/E). The majority of sequences included within the two sister clades D3.2 and D3.3 were assigned to RF-A. The non-RF-A sequences within the D3.2 subclade were assigned to RFs J, L, M, S, W, and X. The D3.3 subclade included 10 RFs (A, F, G, H, J, K, N, R, U, and V) and exhibited the highest richness. The non-RF-A sequences were scattered within the D3 subclade, a phylogenetic pattern indicating independent emergence over time from the circulating RF-A genomes.

### 3.5. Evolutionary History and Geographic Distribution of CVA6

The evolutionary rate was assessed to 4.42 × 10^−3^ nucleotide substitutions per site per year from the CVA6 time-scaled phylogeny inferred with the P1 sequence dataset (Figure 4). The topology confirmed that clades D1–D3 were monophyletic (posterior probability (pp) density = 1). The time to the most recent common ancestor (tMRCA) of subclades D3.2 and D3.3 (tree node N10) was assessed to 2004 (range, 2003–2006; pp = 0.99). The transmission of clades D1 (N7, pp = 0.89) and D3 (N9, pp = 0.99) began in 1995 (range, 1991–1998), 5 years earlier than the transmission of clade D2 (N5, pp = 0.99; range, 1995–2004). The geographic spread of CVA6 was reconstructed with the NextStrain pipeline across 17 countries, using the nucleotide sequences derived from the P1 region (Figure 5). The D3 clade contained CVA6 sequences from Europe and Asia, and showed a high geographic diversity in agreement with the high number of sequences. The sequences determined in the French paediatric population were scattered throughout the D3 clade, and interspersed with those recovered in other countries, suggesting frequent transmission between countries. The phylogenetic relationships between the D3 sequences collected between 2008 and 2018, suggested fast virus migration, allowing sequences sampled in one continent to have ancestors in another.

## 4. Discussion

The present study of the largest European paediatric case series (981 children) shows that atypical forms of HFMD were the most frequent clinical manifestations of CVA6 infections and that most infection cases (estimated proportion, 77%) were caused by the CVA6 recombinant lineages D3/A and D3/H. The remaining cases were caused by a large array of nine other recombinant lineages, five of which were RFs previously unreported before this study. We investigated 245 clinical specimens selected to be representative of the different clinical phenotypes and of the yearly frequency of CVA6 infection cases. Owing to the optimisation of whole-genome amplification and implementation of sequencing of long amplicons with the PacBio^®^ 3rd generation technology, we determined CVA6 full-length genomes in 213 (87%) of the clinical specimens tested. Whole-genome amplification was positive in specimens with Ct values up to 37.8 cycles (determined with a commercial diagnostic test), but significantly decreased beyond 30 cycles. The viral loads were not a limiting factor for our method, because they were high in throat and mouth swabs, which are therefore well adapted to investigate EV genomes associated with HFMD.

Based on our phylogenetic analysis of the structural genomic region, clade D emerged between 1982 and 1995 and diverged into three subclades over a short period (1991–2003), a pattern indicating genetic diversification during geographic virus dispersal. The evolutionary rate determined in our study for the CVA6 structural genomic region was similar to a previous assessment [23]. From phylogenetic data, we assessed that subclade D3 arose in 1995 but active transmission began later between 2003 and 2006, 3–6 years before the HFMD outbreak in Finland [1]. The majority of CVA6 reported in our children population belonged to subclade D3 and a minority to subclade D1. The topology of time-scaled phylogenies indicated that viruses from various geographic origins infected children over time. This is exemplified by virus diversity in various monophyletic recombinant lineages showing close genetic relationships between sequences from France and other countries: D3/N (Australia; [38]), D3/H (United Kingdom, Spain, Denmark; [21]), D3/F (Thailand, United Kingdom, Denmark, Spain, Germany; [22]), and D1/B,G,T (Japan, Australia; [39,40]). The genetic diversity of CVA6 clade D is reminiscent of the pattern of EV-A71 clade C, which diverged into subclades C1 to C5 [41]. However, none of the EV-A71 subclades reached the global predominance of CVA6/D3.

Another difference with EV-A71 is the high number of RFs arising among CVA6 clade D3 and co-circulating over time, while RFs were scarcely reported during transmission of EV-A71 subgenogroups [42]. This pattern suggests differences in surveillance between the two viruses, variations in recombination frequency or in extinction rates of RFs. From our phylogeny based on the 3Dpol gene, the global set of CVA6 genomes consisted of 23 RFs, 16 of them corresponding to the previously reported RFs A–Q [21,22,23]. RF-A became dominant with the worldwide spread of the D3/A CVA6 lineage, the most frequent lineage in our paediatric population (estimated proportion, 58%). This transmission pattern may indicate an advantage for the association of the D3 capsid region with the RF-A non-structural proteins [43]. With an estimated proportion of 19%, RF-H was the second most frequent RF in our patient population. RF-H arose clearly from a recombination event involving a D3 genome and was reported in four European countries between 2012 and 2018. RF-R was identified in only two sporadic CVA6 infection cases and an isolate from Turkmenistan [37]. We also detected this RF in the epidemic multi-recombinant C1-like EV-A71 reported for the first time in Germany [44] and in other EV types including CVA2 from Russia [37], CVA4 from Australia and China [38,45], and CVA10 from India (accession number MH118054). Our data are consistent with a study indicating that EV-A71 and CVA6 have pivotal roles in the generation of intertypic recombination dynamics because their genomes are frequent partners of other species A enteroviruses [46]. However, our knowledge of the true genome diversity is still fragmented and highly biased toward a limited number of enteroviruses of medical interest including EV-A71 and CVA6, leaving out the other (sero)genotypes. Accordingly, with the genomic sequence data available for our study it was impossible to determine the exact origins of the other CVA6 RFs and if they arose through intertypic recombination.

The precise factors involved in the global geographic spread of CVA6/D3 since the mid-2000s are unknown. It was hypothesized that genetic recombination was associated with increased virulence or virus transmissibility of poliovirus vaccine strains or EV-A71 subgenogroup C4a [43]. Pons-Salort and collaborators suggested that an increase in pathogenicity is the epidemiological model that best explains the change in the circulation pattern of CVA6 in Japan since 2010 [47]. In conclusion, the question of a link between CVA6 genetic features and a change in pathogenicity remains open.

## Figures and Tables

**Figure 1 viruses-14-01078-f001:**
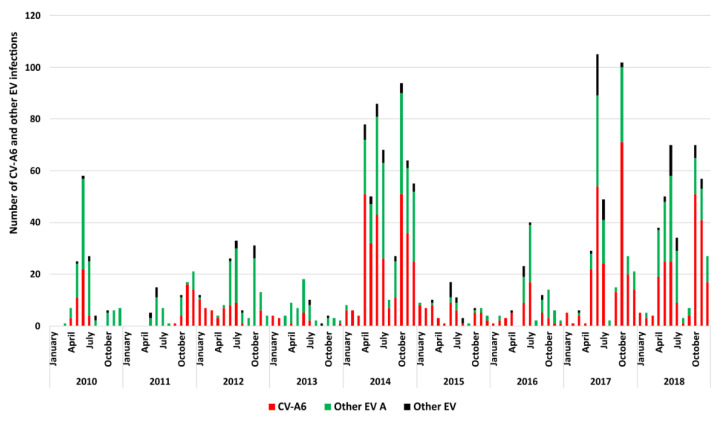
Distribution of CVA6 and other EV types during ambulatory hand-foot-and-mouth disease surveillance. From April 2010 to March 2014, the data are from the surveillance of the population of the Clermont-Ferrand city area. From April 2015 to April 2017, the comprehensive national surveillance was interrupted. Colour code: red, CVA6; green, EV-A other than CVA6; black, other EV types.

**Figure 2 viruses-14-01078-f002:**
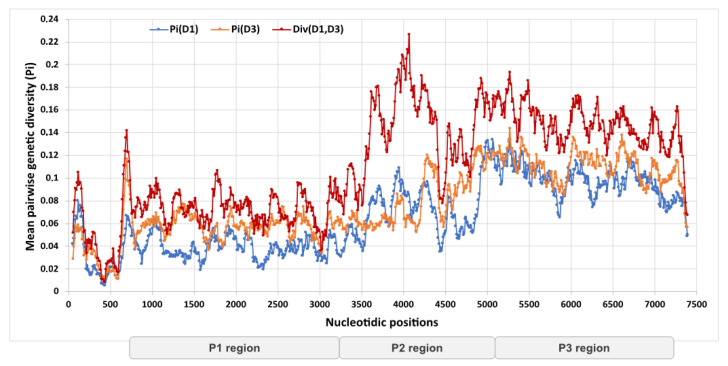
Genetic diversity among the coxsackievirus A6 genomes determined in this study. The mean pairwise genetic diversity was calculated between the 213 CVA6 complete genomes assigned to subclades D1 and D3. The intra-clade nucleotide diversity (Pi) and inter-clade divergence (Div) were calculated as a mean proportion using a sliding window of 100 nts with a step size of 10 nts. Locations of genomic regions in the alignment: P1, nucleotide positions 748–3357; P2, nucleotide positions 3358–5349; P3, nucleotide positions 5350–7351.

**Figure 3 viruses-14-01078-f003:**
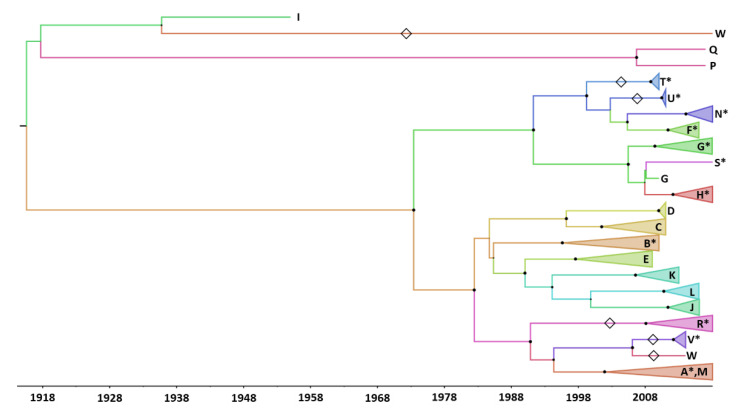
Genetic pool of recombinant forms (RFs) identified among coxsackievirus A6 genomes. The 3Dpol gene was derived from a dataset of 760 (near full) genomes including the 213 CVA6 genomes determined in this study, and was analysed with a Bayesian method. Large full black circles at nodes indicate posterior probability values > 0.9 representing distinct RFs. The clades designated with letters A to Q correspond to the previously identified RFs. Open diamonds along branches indicate RFs previously unreported before this study. The 11 clades labelled with an asterisk include sequences reported in this study. For more clarity, the sequences within each clade were collapsed.

**Figure 4 viruses-14-01078-f004:**
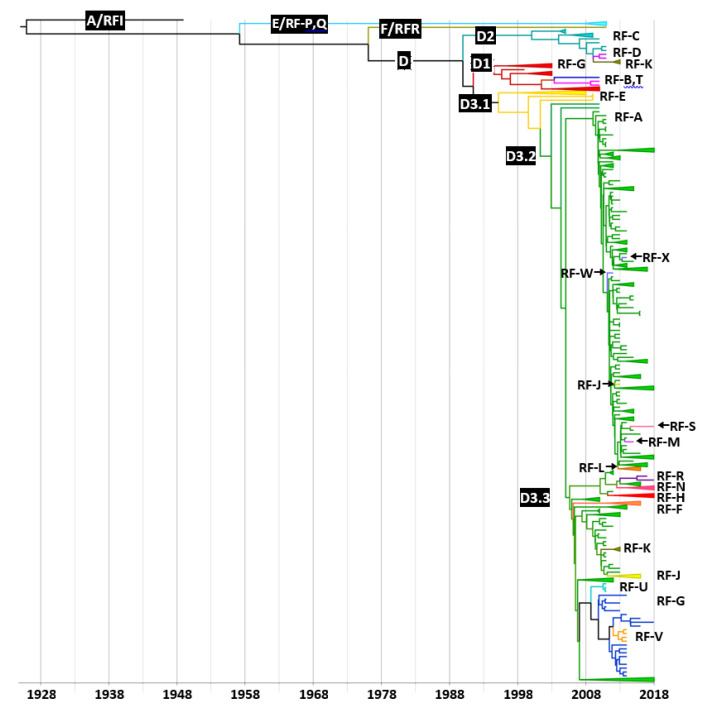
Phylogeny inferred from the P1 genomic region of coxsackievirus A6. A set of 760 P1 sequences was constructed from the 213 complete genomes from this study and 547 genomes available in GenBank. The phylogenetic analysis was performed with BEAST software [30]. Posterior probability is indicated by the size and intensity of the red colour of the circles at the different nodes. The inset table indicate the time to the most recent common ancestor (tMRCA) and the 95% highest probability density interval (95% HPD).

**Figure 5 viruses-14-01078-f005:**
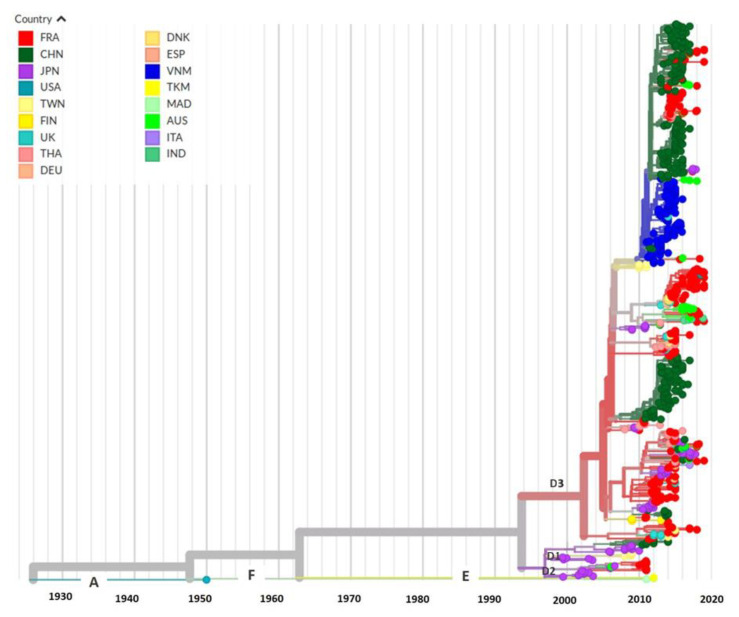
Chronogram and geographic distribution of CVA6 strains. Phylogenetic analysis was performed using the Nextstrain Augur toolkit [34].

**Table 1 viruses-14-01078-t001:** Demographic and clinical data of patients with hand-foot-and-mouth disease/herpangina and genotyping test positive for coxsackievirus A6.

Patient Features ^a^	All CVA6 Infections (*n* = 981)	2010 (*n* = 40)	2011 (*n* = 35)	2012 (*n* = 57)	2013 (*n* = 16)	2014 (*n* = 298)	2015 (*n* = 55)	2016 (*n* = 47)	2017 (*n* = 229)	2018 (*n* = 204)	*p* Value
Age (year) ^b^	1.52 (1.05–2.21)	1.76 (1.17–2.72)	1.80 (1.25–2.38)	1.58 (1–2.24)	2.27 (1.33–3.48)	1.56 (1.05–2.19)	1.47 (1–1.95)	1.51 (1.17–2.34)	1.48 (1.03–2.03)	1.46 (1–2.2)	0.14
Female	408 (42)	10 (25)	19 (54.3)	21 (36.8)	8 (50)	139 (47)	18 (34)	15 (32.6)	83 (36.7)	95 (46.8)	0.02
Male	564 (58)	30 (75)	16 (45.7)	36 (63.2)	8 (50)	157 (53)	35 (66)	31 (67.4)	143 (63.3)	108 (53.2)	
Herpangina	57 (6)	5 (12.5)	4 (11.8)	5 (8.8)	1 (6.2)	24 (8.1)	1 (1.8)	2 (4.3)	11 (4.8)	4 (2.0)	<0.001
Typical HFMD	214 (22)	16 (40)	8 (23.5)	23 (40.3)	3 (18.8)	79 (26.7)	18 (32.7)	3 (6.4)	31 (13.7)	33 (16.2)	
Atypical HFMD	705 (72)	19 (47.5)	22 (64.7)	29 (50.9)	12 (75)	193 (65.2)	36 (65.5)	42 (89.3)	185 (81.5)	167 (81.8)	

^a^ Gender was available for 972/981 (99%) patients and clinical data were available for 976/981 (99%) patients. ^b^ Data are reported as *n* (%) and median [interquartile range].

**Table 2 viruses-14-01078-t002:** Coxsackievirus A6 whole-genome sequencing data.

Sequencing Data	Library 1	Library 2	Library 3
**Generation of circular consensus sequences (CCS)**			
Number of CCS reads	505,649	521,730	665,105
Number of « subreads »	4,571,660	3,608,692	5,319,387
Median number of « subreads » per read	9	6.9	8
**Results after filtering the subreads**			
Number of CCS	201,359	162,812	274,118
Number of « subreads »	3,787,027	2,790,358	4,104,179
Mean number of « subreads » per CCS	18.8	17.1	15
**Results after demultiplexing**			
Number of CCS assigned to the barcodes attributed to samples	108,510	88,608	171,991
Mean number of CCS per barcode	1790	1748	1864
Number of viral genomes obtained	95	95	96
Mean number of nucleotide sequences per sample	1066	865	1709
**Analysed samples**			
Coxsackievirus A6	81	96	49
Enterovirus A71 ^a^	15	-	-
Other types or control ^a^	-	-	47

^a^ Samples containing enterovirus types other than coxsackievirus A6, which were used as controls in the production of libraries.

**Table 3 viruses-14-01078-t003:** Yearly distribution of recombinant forms (RFs) among coxsackievirus A6 complete genomes associated with hand-foot-and-mouth disease/herpangina in France.

RF	Number of Viral Genomes (*n* = 213)	2010 (*n* = 19)	2011 (*n* = 9)	2012 (*n* = 15)	2013 (*n* = 4)	2014 (*n* = 86)	2015 (*n* = 21)	2016 (*n* = 12)	2017 (*n* = 18)	2018 (*n* = 29)	Recombinant Lineage (Clade/RF)
A	123 (58)	9 (47.4)	6 (66.7)	15 (100)	3 (75)	58 (67.4)	14 (67)	3 (25)	8 (44.4)	7 (24.1)	D3/A
B	7 (3.3)	7 (36.8)	-	-	-	-	-	-	-	-	D1/B
C	0	-	-	-	-	-	-	-	-	-	D2/C
D	0	-	-	-	-	-	-	-	-	-	D2/D
E	0	-	-	-	-	-	-	-	-	-	D3/E
F	11 (5.2)	-	-	-	-	9 (10.5)	1 (4)	1 (8.3)	-	-	D3/F
G	13 (6.1)	1 (5.3)	-	-	1 (25)	8 (9.3)	-	2 (16.7)	-	1 (3.4)	D1G; D3/G
H	41 (19)	-	-	-	-	7 (8.1)	6 (29)	6 (50)	8 (44.4)	14 (48.3)	D3/H
I	0	-	-	-	-	-	-	-	-	-	A/I
J	0	-	-	-	-	-	-	-	-	-	D3/J
K	0	-	-	-	-	-	-	-	-	-	D2K; D3/K
L	0	-	-	-	-	-	-	-	-	-	D3/L
M	0	-	-	-	-	-	-	-	-	-	D3/M
N	6 (2.8)	-	-	-	-	-	-	-	1 (5.6)	5 (17.4)	D3/N
P	0	-	-	-	-	-	-	-	-	-	E/P
Q	0	-	-	-	-	-	-	-	-	-	E/Q
R	2 (0.9)	-	-	-	-	-	-	-	1 (5.6)	1 (3.4)	D3/R; F/R
S	1 (0.5)	-	-	-	-	-	-	-	-	1 (3.4)	D3/S
T	2 (0.9)	2 (10.5)	-	-	-	-	-	-	-	-	D1/T
U	3 (1.4)	-	3 (33.3)	-	-	-	-	-	-	-	D3/U
V	4 (1.9)	-	-	-	-	4 (4.7)	-	-	-	-	D3/V
W	0	-	-	-	-	-	-	-	-	-	D3/W
X	0	-	-	-	-	-	-	-	-	-	D3/X

## Data Availability

The data presented in this study are openly available in NCBI GenBank, accession numbers MT814404–MT814616.

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
