# Peer review of "Coxsackievirus A6 Recombinant Subclades D3/A and D3/H Were Predominant in Hand-Foot-And-Mouth Disease Outbreaks in the Paediatric Population, France, 2010–2018"

_viruses, 2022, doi:10.3390/v14051078_

Round 1

Reviewer 1 Report

This manuscript is well written. Data collection, computational analyses and interpretation are all excellent. Extensive classification towards various RFs is done for some known RFs as well as some novel RFs. In particular in Discussion the authors pointed out the analogy between CVA6 and EVA71 in terms of their evolution and recombination. They also clearly discussed some shortcomings regarding data collection bias in trying to draw conclusion about these recombinants. I truly feel that readers who are interested in understanding/investigating genomic recombination of Enteroviruses would benefit from this manuscript. I only have some very minor comments: (1) In line #168, the authors stated using window size 200 and step size 20 in scanning the alignment. I wonder why previously used 250 and 25 as outlined in line#160 is not used here. (2) Table 3 lists RFs found in this study (n=213), including A to X except O and Z. In Figure 3 (n=760), however, I noticed clade 'O' near the upper-right corner, yet did not see clade 'Q'. Is this a mis-labeling? (3) Figure 4 legend: a minor mistake was spotted in line #312, where 557 genomes should be 547, I think.  

Author Response

We thank the reviewer for the positive appreciation of our manuscript and the comments.

(1) In line #168, the authors stated using window size 200 and step size 20 in scanning the alignment. I wonder why previously used 250 and 25 as outlined in line#160 is not used here.

Correct, this is a mistake. We have done various analyses and in Figure 2, we present the results obtained with the parameters indicated in the legend to the figure: window = 100 nts, step size = 10 nts. We used these parameters to report the variations in genetic diversity between genomes to the finest details.

(2) Table 3 lists RFs found in this study (n=213), including A to X except O and Z. In Figure 3 (n=760), however, I noticed clade 'O' near the upper-right corner, yet did not see clade 'Q'. Is this a mis-labeling?

Correct. The letter ‘O’ has not been not used in the referencing of RFs, accordingly ‘O’ should be ‘Q’. Mis-labelling corrected in the figure.

(3) Figure 4 legend: a minor mistake was spotted in line #312, where 557 genomes should be 547, I think.

Correct, done.

Author Response

We thank the reviewer for the positive appreciation of our study.